# Interplay and Dynamics of Chromatin Architecture and DNA Damage Response: An Overview

**DOI:** 10.3390/cancers17060949

**Published:** 2025-03-11

**Authors:** Susanna Ambrosio, Anna Noviello, Giovanni Di Fusco, Francesca Gorini, Anna Piscone, Stefano Amente, Barbara Majello

**Affiliations:** 1Department of Biology, University of Naples “Federico II”, 80126 Naples, Italy; anna.noviello@unina.it (A.N.); g.difusco@studenti.unina.it (G.D.F.); 2Department of Molecular Medicine and Medical Biotechnologies, University of Naples “Federico II”, 80131 Naples, Italy; francesca.gorini@unina.it (F.G.); anna.piscone@unina.it (A.P.); stamente@unina.it (S.A.)

**Keywords:** chromatin architecture, double-strand breaks, genome stability

## Abstract

This review explores the intricate relationship between DNA double-strand break (DSB) repair and chromatin architecture, examining how chromatin influences DSB repair at multiple levels, from histone modifications to nuclear organization, while highlighting the emerging complexity of this interplay.

## 1. Introduction

Beyond compaction, replication, and genome segregation, chromatin folding is essential for regulating gene expression, maintaining epigenetic memory, and repairing DNA damage. Mammalian cells experience thousands of endogenous and exogenous DNA-damaging events per day, leading to double-strand breaks (DSBs) that, if left unrepaired, can result in cancer and genetic disorders. However, dedicated molecular mechanisms are activated to preserve DNA integrity.

Since DNA is not merely a sequence of nucleotides but is closely associated with histones and nuclear landmarks, it is not surprising that histone modifications, as well as nuclear architecture, play a pivotal role in shaping chromatin conformation. These structural changes help orchestrate the efficient DNA damage response (DDR) and repair, ensuring genome stability.

Here, we review the main levels of chromatin architecture, histone modifications, and remodeling complexes, as well as nuclear organization, which are intimately involved in the response to and repair of DSBs. We focus on the functional interplay between different nuclear structures in preserving genome stability and the crosstalk with DDR and repair pathways.

## 2. Chromatin Architecture: From Nucleosome to Higher Order Organization

The eukaryotic genome is characterized by a multi-scale hierarchical organization, ranging from simple structures to more complex, higher-order, and non-random forms, in which DNA is folded through a dynamic and well-regulated process that plays both structural and functional roles in cellular homeostasis.

At the most basic level of its organization, chromatin exists as an 11 nm “beads-on-a-string” structure, where 147 base pairs of DNA are tightly wrapped around an octamer composed of two copies each of the four core histones—H2A, H2B, H3, and H4—along with the H1 linker histone [1]. These interactions confer stability while maintaining a dynamic nature due to post-translational modifications (PTMs) and the incorporation of histone variants, which alter chromatin structure and interaction properties [2] (Figure 1A).

Subsequently, nucleosomes are thought to be packaged into a 30 nm chromatin fiber in vitro. However, the exact nature of chromatin fibers in vivo remains debated. Recent observations using cryo-electron microscopy (Cryo-EM) and stochastic optical reconstruction microscopy (STORM) suggest that chromatin fibers typically range from 5 to 24 nm in diameter, forming irregularly folded chains with a zig-zag arrangement. In contrast, other studies propose that chromatin can adopt a solenoid-like structure [3,4] (Figure 1B).

Moreover, high-resolution chromosome conformation data highlight the existence of chromatin loops and long-range contacts, which define the next levels of chromatin organization. Chromatin loops serve multiple functions and exist in a wide range of sizes, from a few kilobases to more than 100 megabases [5]. This higher-order packaging, which involves the folding of chromatin into a series of loops, is catalyzed by ring-shaped protein complexes that bring the enhancers into spatial proximity with their target promoters [6] (Figure 1C).

The organization of the genome at these levels requires multiple factors to maintain and restore chromatin structure throughout the cell cycle, development, and signaling. One of the most studied proteins involved in this process is CTCF, a CCCTC-binding factor and an eleven-zinc-finger (ZF) transcriptional regulator that recognizes numerous DNA motifs through different combinations of its ZFs [7]. Another key player is cohesin, a multi-subunit protein complex belonging to the structural maintenance of chromosomes (SMC) family. Cohesin possesses ATPase activity and forms a ring-like structure that enables DNA passage and facilitates chromatin folding into loops [8,9,10,11].

Hi-C and related chromosome capture techniques have further revealed that interphase chromatin is organized into sub-megabase-scale domains known as topologically associating domains (TADs), where two DNA sequences have a higher probability of interacting with each other than with sequences outside the domain [12] (Figure 1D). While the mechanisms underlying TAD formation remain unclear, the loop extrusion model—originally proposed to describe loop formation during mitosis—may also explain loop formation in interphase. This model suggests that long-range *cis* interactions within a DNA molecule are generated by cohesin, which extrudes small loops that progressively enlarge. When these loops encounter CTCF molecules, which act as boundaries, the extrusion process halts [13,14,15].

TAD boundaries are enriched in transcription start sites, housekeeping genes, and highly transcribed genes. However, single-cell Hi-C and microscopy studies reveal that TADs represent an averaged population-level view, with interactions varying between individual cells [11,16].

Hi-C also easily detected contacts within (cis) and between (trans) chromosomes that can self-interact, thus revealing two types of compartments enriched, respectively, with euchromatin (compartment A) and heterochromatin (compartment B). This phenomenon of chromatin segregation, known as compartmentalization, [14] can lead to the subsequent formation of sub-compartments, each of them characterized by histone modifications and motifs recognized by bridging factors, that also have the ability to phase-separate, creating a micro-environment in which multiple loci can aggregate simultaneously [17] (Figure 1E). A distinct subset of B sub-compartments is tethered to the nuclear lamina (also referred to as the Lamina) during interphase. These lamina-associated domains (LADs) primarily contain transcriptionally silent, compacted heterochromatin enriched with H3K9me2/3 and H3K27me histone marks. Like TADs, LADs are shaped by the epigenetic landscape and represent one of the most dynamic and heterogeneous high-order genome organization features within the nuclear environment [18]. Moreover, lamina proteins are interaction partners of many duplication [19,20], transcription [21], and DNA damage repair factors [22,23,24,25], as their direct mutation, dysregulation, or incorrect post-translational maturation can alter any of these processes leading to various diseases [26]. As well as contributing to the nucleus and chromatin architecture, lamina is also involved in mechanosensing [27], the ability of a cell to sense mechanical stimuli of surrounding microenvironment. The linker of nucleoskeleton and cytoskeleton (LINC) complex spans the inner and outer membranes of the nuclear envelope to structurally support the nucleus and play a role in translating physical cues and alterations into biochemical signals, thereby allowing the cell to adapt its morphology as well as gene expression [28].

## 3. Chromatin Architecture and DNA Damage Response

DNA double-strand breaks (DSBs), in which both strands of the DNA helix are severed, are one of the most harmful lesions known to occur in genomes. To counteract these threats, cells employ a variety of DDR pathways, including cell-cycle checkpoints, programmed apoptosis, and direct DNA repair mechanisms [29]. Non-homologous end-joining (NHEJ) and homologous recombination (HR) are the two main DSB repair pathways: which one the cell opts for depends on the cell-cycle phase and the extent of DNA-end resection [30]. NHEJ operates ubiquitously throughout the cell cycle, requires limited resection and involves simple ligation of the two broken ends with little to no homology, but it often results in small nucleotide insertions or deletions at the repaired site [31]. NHEJ is initiated by the binding of Ku70-Ku80 heterodimer, which subsequently recruits DNA-PKcs to promote the synapsis of loose DNA ends [32]. Final ligation is carried out by DNA ligase 4 (LIG4) in conjunction with XRCC4. Importantly, 53BP1 prevents extensive end resection during the G1 phase [33]. In contrast, HR uses an intact copy of the damaged locus as a template and requires 5′ end resection, which generates a single-stranded 3′ DNA end that invades and perfectly copies a homologous sequence to safely repair the DSB. This requires the action of RAD51, which assembles nucleoprotein filaments with the 3’ ssDNA ends and allows synthesis-dependent strand annealing from the sister chromatid [34]. RAD51 loading onto ssDNA ends is forwarded by BRCA1, which repositions 53BP1, allowing end-resection and thus shifting the choice in favor of HR [35]. Consequently, to avoid the use of the homologous chromosome as a template in G1 and loss of heterozygosity, canonical HR is constrained to the S/G2 phase only [30], while non-canonical forms of HR may occur also in the G1 phase. If both these pathways are compromised (such as in cancer cells, depending on their specific background and acquired mutations, or in cells experiencing replicative stress and extensive DNA damage that overwhelms the main repair mechanisms), alternative end joining (alt-EJ) and single strand annealing (SSA) can serve as backup routes, albeit at the cost of being highly mutagenic [36].

This elaborate genome surveillance network relies on continuous interactions between DNA repair mechanisms and spatial genome organization. Here, we summarize the latest findings on chromatin dynamics in response to DSBs and how they influence the DDR at both local and global levels (Figure 2).

### 3.1. Local Chromatin Reorganization upon DSBs

#### 3.1.1. DSBs Histone Code

Nucleosome modifications regulate DNA accessibility by governing stiffness, flexibility, and mobility (Figure 3A). They also directly contribute to the recruitment of DSB repair factors, repair pathway choice, and the recruitment of molecular machines to balance repair with transcriptional suppression, ensuring that the two processes do not interfere.

Upon DNA damage, the DDR initiates a cascade of events to detect, signal, and repair the damage. DSBs are detected by key sensors, such as the MRN complex (MRE11, RAD50, NBS1), that recognize DSBs and recruit the ATM kinase, which is subsequently activated by autophosphorylation [37,38]. ATM phosphorylates histone H2AX at serine 139, forming γH2AX, that spreads along the chromatin [39], creating a platform for the recruitment of the mediator of DNA damage checkpoint protein 1 (MDC1). MDC1 acts as an amplifier of MRN/ATM complex activity and propagates γH2AX up to 2 megabase pairs, in cis, from the DSB [40]. γH2AX recruits RNF8-UBC13, which promotes K63-linked polyubiquitination of H1. This modification is then recognized by RNF168, which further extends the K63-linked polyubiquitin chains and monoubiquitinates H2A at K13/15, generating a DSB-flanking platform for repair factors such as 53BP1 and the BRCA1/BARD1 complex [41,42].

Notably, the balance between ubiquitination and deubiquitination ensures that chromatin is dynamically remodeled for efficient repair while preventing excessive or misdirected repair activity [43].

Acetylation is associated with chromatin relaxation, which allows repair factors to access DNA lesions. In humans, two HATs are mainly responsible for this phenomenon following DSB induction: hMOF, a member of the MYST family of HATs, which acetylates histone H4 at lysine 16 (H4K16ac) [44], and KAT5/TIP60, which acetylates two different histones, H4 at lysine 16 (H4K16) and H2A at lysine K15 (H2AK15) after DNA damage [45]. Notably, H4K16 acetylation has been linked to negatively regulating 53BP1 binding, a critical determinant in the repair pathway choice between HR and NHEJ [46,47]. H4K16 acetylation provides a steric hindrance to 53BP1 for its binding to the adjacent H4K20me2 [47]. It is worth noting that genomic regions characterized by high transcriptional activity are associated with higher levels of H4K16ac compared to gene-poor chromosomal regions. This results in a preferential recruitment of HR-related DSB repair proteins [48]. On the other hand, deacetylation of certain residues, such as H3K56ac, by the histone deacetylase SIRT6, and H4K16ac, by SIRT1, is necessary for the efficient completion of DNA repair processes [49,50,51].

Methylation marks can act as binding sites for reader proteins that facilitate the assembly of repair complexes. For example, H3K36me3 promotes HR repair by recruiting proteins like RAD51 and CtIP, which are crucial for DNA end resection and strand invasion [52]. Additionally, BARD1 interacts with H3K9me2 in response to DNA damage and retains the BRCA1 complex to promote HR, while inhibiting NHEJ [53]. Recently, it has been demonstrated that H4K20 methylation levels oscillate during the cell cycle and contribute to balance the repair pathway choices [54,55]. In pre-replicative cells, H4K20me1/2 is recognized by 53BP1, whereas newly incorporated H4K20me0 in the S phase recruits BRCA1/BARD1 to DSB sites, displacing 53BP1 and inhibiting NHEJ repair [54,55]. Moreover, RNF8 and RNF168 indirectly regulate 53BP1 accessibility to methylated histones, by ubiquitinating the H4K20me2-binding factor L3MBTL1 triggering its dissociation from chromatin and mediating the degradation of JMJD2A, enhancing the 53BP1–H4K20me2 interaction at the damaged site [56,57].

Finally, nucleosome remodelers play a critical role in local chromatin relaxation during DSB repair by mediating nucleosome eviction, editing, and positioning [58]. A crucial step in the choice of DSB repair pathways is DNA end resection. HR requires extensive end resection to generate single-stranded DNA tails capable of invading the homologous strand, but nucleosomes present a barrier to resection nucleases [59]. BAF and PBAF, both part of the SWI/SNF family, as well as the INO80 complex, have been implicated in nucleosome eviction during DSB repair and resection, underscoring their essential role in HR repair [60,61].

#### 3.1.2. Transcription Regulation at DSBs

When DSBs occur, transcriptional silencing in the surrounding regions prevents collisions between transcription and repair machinery. This silencing is achieved through the transient establishment of a repressive chromatin landscape and the direct eviction of the RNA polymerase II (RNAPII) complex from the damaged site [62].

DDR kinases and poly-ADP-ribose polymerase 1 (PARP1) signaling orchestrate transcriptional shutdown upon DNA damage. ATM activates polycomb repressive complexes (PRCs) that ubiquitinate histone H2A at lysine 119 [63]. This modification spreads across kilobases, blocking transcription elongation in regions surrounding the break. ATM also phosphorylates ENL/AF9, a subunit of the super-elongation complex, reinforcing the transcriptional blockade [64]. PARP1 inhibits RNAPII activity by directly binding to and modifying the positive elongation factor b (P-TEFb), preventing RNAPII hyperphosphorylation [65]. Additionally, PAR chain formation on RNAPII recruits the negative elongation factor (NELF), further impairing RNAPII elongation [66].

While coding transcription halts, non-coding transcription can be initiated at DSB ends, and these events have been shown to profoundly affect the efficiency and accuracy of DSB repair [67,68]. DICER- and DROSHA-dependent noncoding RNAs, named DNA damage response RNAs (DDRNAs), and their precursors, damage-induced long noncoding RNAs (dilncRNAs, a category of non-coding RNAs synthesized at DSBs by a fully assembled RNA polymerase II preinitiation complex, in conjunction with mediator complex MED1 and elongation factor CDK9) are synthesized at damage sites [67,69]. These RNAs help the secondary recruitment of DDR and scaffold the repair complexes favoring liquid–liquid phase separation (LLPS) of the DDR foci component 53BP1 [70,71,72]. Following a DSB, the negative charges due to PARylation by PARP enzymes recruit RNA binding FET family proteins, which present highly disordered structural domains that facilitate the formation of liquid-phase compartments [73]. These liquid-like droplets concentrate repair factors, isolate damaged DNA, and provide a microenvironment conducive to efficient repair, dynamically assembling and disassembling in response to the progression of the repair process [74]. Pessina et al. described the LLPS properties of 53BP1 condensates, demonstrating that dilncRNAs are necessary to stimulate the LLPS of 53BP1 and other DDR factors into discrete foci at DSB sites [72] (Figure 3B).

In transcription-coupled repair, RNA transcripts may even act as templates for HR, forming DNA:RNA hybrids during the G0/G1 phase in non-replicating cells, where the undamaged sister chromatid is unavailable as a donor template; thus, canonical HR is unattainable [75,76].

The DDR also involves dynamic changes in chromatin architecture to accommodate repair factors and prevent aberrant transcription. The ISWI family of remodelers, including Remodeling and Spacing Factor 1 (RSF1) and Williams–Beuren syndrome transcription factor (WSTF), regulate preexisting histone modifications in transcribed chromatin, ensuring efficient repair and precise transcriptional control. Upon DNA damage, RSF1 interacts with HDAC1 to deacetylate H2AX at lysine 118, a marker of active transcription, facilitating subsequent ubiquitination at lysine 119. WSTF modulates H2AX phosphorylation states to balance transcriptional repression during repair and activation afterward. Upon DNA damage, H2AX undergoes a phosphorylation switch from pre-existing H2AX-pY142 to γH2AX, disrupting its interaction with RNAPII and repressing local transcription. During DNA repair, WSTF re-phosphorylates H2AX-Y142, enabling transcription-coupled HR (TC-HR) in the G1 phase.

Ultimately, chromatin remodeling ensures that transcriptional silencing is temporary and reversible. Once repair concludes, chromatin modifiers restore histone marks to reactivate transcription, allowing cells to protect genome stability while maintaining cellular homeostasis.

### 3.2. Large-Scale Chromatin Reorganization upon DSBs

#### 3.2.1. TADs Lead DDR Chromatin Domains’ Establishment

Immediately after a break, γH2AX domains spread across neighboring chromatin regions, marking extensive areas around the DSB. These domains stabilize broken chromosomes until repair occurs and serve as recruitment platforms for repair factors [77]. However, the observed γH2AX profiles, which are often asymmetric, show gaps, and exhibit variable levels of modification within the domain, do not align with the traditional processive model of propagation along the chromosome [78]. High-resolution microscopy further reveals that γH2AX foci are not continuous but instead form spatially distinct nano-domains around DSB sites [79,80].

The interactome at the break site determines the extent and density of γH2AX modification, which is primarily, though not entirely, confined to TADs [81] (Figure 3C). When a DSB disrupts a TAD boundary, γH2AX domains can expand into adjacent TADs. In contrast, DSBs near TAD boundaries generate asymmetric γH2AX domains, potentially influencing repair efficiency [81]. Consistently, an increase in TAD boundary strength has been observed after exposure to ionizing radiation [82].

It has been proposed that this process is driven by loop extrusion facilitated by cohesins [83]. On both sides of the DSB, divergent one-sided loop extrusion promotes the bidirectional spread of γH2AX. ATM, recruited near the DSB, phosphorylates H2AX on nucleosomes as they are extruded by cohesins [83]. As an architectural factor, cohesins play a key role in the DDR by acting as physical barriers, limiting interactions with distal chromatin regions and confining the repair process to the vicinity of the damage, thereby minimizing ectopic recombination and maintaining repair fidelity [83,84].

As extrusion progresses, γH2AX-modified chromatin expands but halts at boundary elements such as CTCF-bound loci that define TAD borders [79,81]. CTCF plays a pivotal role in organizing and clustering γH2AX foci spatially. Deleting a CTCF motif reduces γH2AX spread within a TAD [81], and cells with reduced CTCF levels exhibit fewer and smaller γH2AX foci containing less DNA compared to controls; this disruption results in delayed DDR activation and reduced repair efficiency [79].

#### 3.2.2. DNA Repair Is Dependent on Preexisting Chromatin Features

Eukaryotic nuclei can be divided into two main configurations: euchromatin, which is more open and actively transcribed and contains most of the protein-coding genes, and heterochromatin, which is more condensed and generally silenced. These structural and functional differences between the two forms of chromatin have fundamental implications for the DNA damage response, including the repair pathway choice.

Heterochromatic regions, being more compact, present limited access to DNA repair factors, making the repair of DSBs more challenging. Heterochromatic regions are typically associated with repetitive sequences and enriched for histone H3K9 trimethylation (H3K9me3), which acts as a signal for transcriptional silencing and the formation of stable heterochromatin domains [85]. Additionally, the HP1 protein (Heterochromatin Protein 1, including HP1α, HP1β, and HP1γ), which binds to H3K9me3-rich regions, plays a crucial role in maintaining the heterochromatin high-order organization [86].

Chromatin decompaction is required for efficient DNA repair in heterochromatin. This reorganization requires the displacement of core components, such as HP1β and KRAB- associated protein-1 (KAP1), allowing repair factors to access the DSBs, facilitating chromatin rearrangement, and aiding in the repair of the damage [87,88]. Specifically, ATM-dependent KAP-1 phosphorylation alters KAP-1 affinity for chromatin and facilitates heterochromatin remodeling in DDR [87], while CK2-mediated phosphorylation regulates the mobilization of HP1-β from chromatin after DNA damage [88].

The abundance of repetitive sequences in heterochromatin makes these regions highly susceptible to illegitimate recombination following the formation of DSBs. Besides local relaxation, DSBs in heterochromatin are also subject to spatial rearrangement within the nucleus. Specifically, DSBs in heterochromatic regions tend to move towards the periphery for the completion of recombinational repair, which is believed to reduce the risk of improper recombination between pericentromeric repeats and favor more accurate repair [89,90,91,92]. In *Drosophila*, this movement is mediated by recruitment of the SMC5/6 complex and the associated SUMO E3 ligase/Nse2 homolog, which interact with DNA and mediate the SUMO-dependent relocation of heterochromatic DSB away from regions with high-density repetitive sequences [93,94]. Damaged heterochromatic regions can protrude into euchromatin, facilitating repair. However, this mobilization can occasionally lead to clustering of DSBs, increasing the risk of chromosomal translocations. Thus, while movement enhances repair efficiency, excessive clustering can contribute to genomic instability [95,96]. Remarkably, heterochromatin-specific histone modifications and transcriptional silencing are retained during this process, indicating that, despite massive decompaction, the identity of heterochromatin is preserved [79,92,97,98].

Repair pathway choices differ among heterochromatin domains, reflecting a compartmentalization of DNA repair in the eukaryotic nucleus, and suggesting that spatial positioning of the DSB inside the nucleus may also play a role [99,100]. In mouse cells, centromeric DSBs recruit RAD51 throughout interphase, while pericentromeric DSBs use HR only in S/G2 [92]. Telomeric DSBs are exclusively repaired by HR and alt-EJ [101]. LADs are repaired via error-prone NHEJ and alt-EJ [102,103], while nucleolar DSBs utilize both NHEJ and HR [104,105,106]. Thus, significant differences have been identified in how DSBs are processed within distinct silenced chromatin compartments. Further research is required to fully elucidate the molecular factors governing this compartmentalization and its impact on genome stability in the eukaryotic nucleus.

DSBs arising in euchromatin are not repaired uniformly as well, and it is now evident that the transcriptional state of chromatin prior to DSB formation critically influences the repair process [107]. By using the DIvA cell line, in which DSBs can be induced at specific genomic sites using the AsiSI endonuclease, Legube group demonstrated that in euchromatin, only transcriptionally active genes are prone to HR repair in G2 [108]. These loci are directed to HR repair via the trimethylated histone H3K36 mark, which is associated with transcriptional elongation. Specifically, RAD51 recruitment at these DSBs depends on the H3K36me3-LEDGF axis. Importantly, damaged transcriptionally active loci are refractory to repair in G1 [109]. Instead, they persist and cluster within foci, a process requiring the MRN and LINC complexes as critical mediators, as well as on the phase-separation properties of 53BP1. This clustering strategy may protect transcribed chromatin from error-prone repair, delaying NHEJ in G1 and promoting HR repair during the S and G2 phases [109]. Other studies, however, have shown the formation of clusters even in G2, suggesting a potentially broader active role for their formation, not strictly confined to a single phase of the cell cycle [110,111].

#### 3.2.3. Nuclear Compartmentalization Following DSBs’ Induction

Recently, two independent groups explored how chromatin compartmentalization influences the cellular response to DSBs, uncovering novel insights into the mechanisms and implications of DSBs’ clustering [112,113].

According to the Legube group, the A and B compartments remain largely stable following damage induction, with a higher frequency of clustering observed among DSBs located within loci of the A compartment [112]. Similarly, the Goutier group reports a comparable trend, which is further amplified by a B-to-A switch occurring in approximately 15% of damaged loci; these loci exhibit enriched interactions with other regions of the A compartment, contributing to the formation of functional repair domains [113].

Arnould et al. identified the formation of a new chromatin compartment, termed the “D compartment”, that emerges after DSBs’ formation in mammalian cells. This compartment is created through the clustering of damaged TADs marked by γH2AX and 53BP1 in a cohesin-independent manner. Its assembly is regulated by ATM kinase activity and predominantly occurs during the G1 phase of the cell cycle [112].

Although LLPS is essential for the early stages of γH2AX foci formation, the mechanism behind the clustering is consistent with polymer–polymer phase separation (PPPS). This phenomenon involves self-interactions among chromatin-bound 53BP1 molecules, promoting their association within damaged TADs while isolating them from neighboring, undamaged chromatin [112] (Figure 3D).

Zagelbaum et al. complement these findings demonstrating that DSB mobility and clustering are driven in part by nuclear actin polymerization, mediated by ARP2/3 and WASP pathways [113].

The Legube group also highlights the role of R-loops, structures formed by hybrid DNA–RNA interactions, in facilitating D-compartment formation and DDR gene recruitment. Genes that accumulate R-loops are more likely to localize to this compartment, and the disruption of R-loop dynamics, such as through RNase H1 overexpression, reduces DSB clustering. Conversely, enhancing R-loop accumulation increases the formation of the D compartment, further linking these structures to chromatin reorganization during DNA damage [112] (Figure 4).

A key finding of this study is the functional role of the D compartment in DDR gene activation [112]. Certain DDR genes, enriched in R-loops, are physically recruited to the D compartment, which optimizes their transcriptional activation after DNA damage [112]. This indicates that the spatial reorganization of chromatin into the D compartment directly supports an effective response to DNA damage. However, this beneficial reorganization comes at a cost: the clustering of damaged regions increases the risk of chromosomal translocations, a hallmark of genomic instability observed in many cancers [95,96,112,113].

Interestingly, pharmacological inhibition of DNA-PK enhances clustering [112,113] while suppressing translocations [113]. This highlights that, in the absence of NHEJ machinery, DSBs remain unrepaired, leading to γH2AX/53BP1 accumulation and enhanced clustering. Additionally, it underscores that chromosome translocations require an end-joining step. This seemingly contradicts previous observations and the nature of DSBs prone to clustering [108,110,112,113], which typically occur at transcribed loci and are preferentially channeled to HR repair. Thus, the induction of DSBs triggers a multiscale alteration of genomic architecture, driving DSBs’ mobility for clustering into homology-driven repair domains, while simultaneously increasing the frequency of NHEJ-mediated translocations.

These findings underline the dual nature of chromatin reorganization in response to DNA damage: while it promotes efficient repair and gene activation, it also elevates the risk of deleterious chromosomal rearrangements. This trade-off between repair efficiency and genomic stability has significant implications for understanding cancer development and the role of chromatin architecture in maintaining genome integrity.

#### 3.2.4. DSBs’ Clustering Depends on the Crosstalk Between Nuclear Structures and DNA Repair

The LINC complex is embedded in the nuclear envelope, and its constituents connect the cytoskeleton with nucleoskeletal proteins also playing a prominent role in DNA damage repair: transluminal interaction between SUN (Sad1-UNC-84 homology) and KASH (Klarsicht, ANC-1, and Syne homology) domain proteins is important for mechanotransduction and chromatin mobility to correctly balance the pathway choice between NHEJ and HR [27]. SUN1 sequesters the Ku70/80 heterodimer, thus inhibiting DNA-PK catalytic activity [114]; facilitates chromatin relocation upon DSB formation involving SUN2 and the phase-separation properties of 53BP1; and promotes RAD51 loading at DNA damaged sites, consequently tipping the balance towards HR rather than NHEJ [115]. In addition, SUN 2 is specifically involved in the clustering of γH2AX foci as its depletion lowers the number of cluster positive cells [109]. Supporting the view that loci are repaired based on their chromatin context, breaks within actively transcribed regions enriched with H3K36me3 histone modification show a tendency to be preserved from unfaithful repair and translocations until end-resection in S/G2 by self-segregation into D compartments. [112] In contrast, inactive genes in heterochromatic LADs show limited displacement after damage and are rapidly repaired in situ by NHEJ factors [102,103]. It has been also reported that SUN1, as well as lamin B1, were upregulated in LMNA-depleted cells and model systems [26], fueling speculation of a functional redundancy that leads to a compensatory mechanism, but this does not occur reciprocally as lamin B1 depletion is not always compensated by overexpression of the other lamins, suggesting a more peculiar role for B-type lamins [116]. The LINC complex has also been associated with damaged telomers roaming within the nucleus that, contrary to that of active loci, is a phenomenon linked with the repair by NHEJ [32]. In a physiological environment with few DNA lesions, this mechanism is very efficient in preserving genome stability, but in the case of a large number of DSBs, this could lead to error-prone repair or translocations, as SUN1/2-deficient cells showed a significant reduction in the mobility of dysfunctional telomeres and their fusion [117]. How the LINC is molecularly involved in the chromatin movement remains to be determined, but one possible speculation hints at the capacity of the nuclear envelope to form invaginations; this could be driven by microtubules’ forces that are transduced by the LINC onto the chromatin as random “poking” of the nucleus, in response to DNA damage [32]. Chromatin mobility is also reliant on other physical constraints in the nucleus: evidence for the nuclear pore complex (NPC) involvement in specialized repair is supported by SUMOylation related anchoring of collapsed replication forks with NPCs. Moreover, Emerin, an integral nuclear membrane protein that binds directly to lamin A/C, modulates nuclear actin polymerization, especially in G2-phase cells undergoing DSBs’ repair, aiming to localize lesions into hubs of HR activity. Actin polymerization requires nucleation, and inhibition of ARP2/3 reduced DSBs’ clustering, while loss of lamin A/C and Emerin affects nuclear myosin 1 (NM1) activity, resulting in enhanced chromatin movement [118].

## 4. Conclusions and Future Perspective

Preserving genome stability is essential for cellular homeostasis. This complex process requires continuous interactions between DNA repair mechanisms and spatial genome organization. Consequently, DNA repair extends beyond classical DDR factors to encompass a broader network that includes chromatin remodeling, replication, transcription, genome organization, the nucleocytoskeleton, and phase separation. These interconnected components synergize to support DDR signaling pathways, enhance the mobility and repair of damaged DNA, and facilitate the dynamic recruitment of DDR factors.

Emerging tools such as single-molecule imaging and Hi-C techniques are beginning to illuminate the relationship between genome organization and DSB repair [79,80,112,113,119,120]. These approaches are gradually revealing the dynamic modifications of damaged chromatin and suggest that chromosomal organization influences not only repair efficiency but also the balance between genome stability and evolutionary adaptability. Integrating these technologies with multiomic datasets could enhance our understanding of tumor evolution, chemoresistance, and metastasis from a multidimensional perspective, with implications for cancer research and clinical applications.

Key cancer features can modulate chromatin dynamics and repair efficiency, potentially influencing DDR and enabling therapy resistance on one hand, while providing potential therapeutic vulnerabilities on the other. Polyploid cells exhibit altered 3D genome organization, affecting DNA accessibility and fostering genomic instability [121,122]. Cancer stem cells (CSCs) often maintain a closed chromatin state, delaying repair and enhancing resistance to genotoxic stress [123]. Additionally, cells with DDR deficiencies, such as BRCA1/2-mutant tumors, may reorganize their 3D genome in response to DNA damage to compensate for repair defects and suit their unique needs [124].

Over the past decade, targeting key DDR components has emerged as a promising strategy for developing cancer therapies. The introduction of PARP inhibitors, the first class of anticancer drugs exploiting synthetic lethality, has already reached clinical application, particularly in the treatment of BRCA-mutated ovarian, breast, pancreatic, and prostate cancers [125]. Additionally, numerous DDR-targeting drugs, such as ataxia-telangiectasia and Rad3-related (ATR) kinase inhibitors (e.g., berzosertib), ATM inhibitors (e.g., AZD0156), DNA-PK inhibitors (e.g., VX-984), and checkpoint kinase 1 and 2 (CHK1/CHK2) inhibitors (e.g., prexasertib), are undergoing various stages of clinical trials [30,125]. Furthermore, compounds that inhibit specific histone-modifying enzymes, such as HDAC inhibitors (e.g., vorinostat, romidepsin), have also shown promise in cancer treatment and are being explored for non-cancer-related conditions [126].

If we begin to consider chromatin dynamics as an integral part of DDR, investigating their potential as a therapeutic target for patient benefit represents an exciting research avenue. So far, curaxin-CBL0137 has been reported to target the 3D genome organization by interacting with CTCF-binding sites, thereby affecting long-range cis-regulatory elements [127]. Meanwhile, the potential effects of epigenetic drugs on 3D chromatin architecture are being investigated as a strategy to overcome cancer therapy resistance [128].

The understanding of the intricate relationship between chromatin organization and DNA repair, and how these processes influence cellular and disease mechanisms, is still in its early stages. By integrating current knowledge, we encourage further research into the potential of targeting chromatin organization and dynamics in the context of genome stability and cancer therapy.

## Figures and Tables

**Figure 1 cancers-17-00949-f001:**
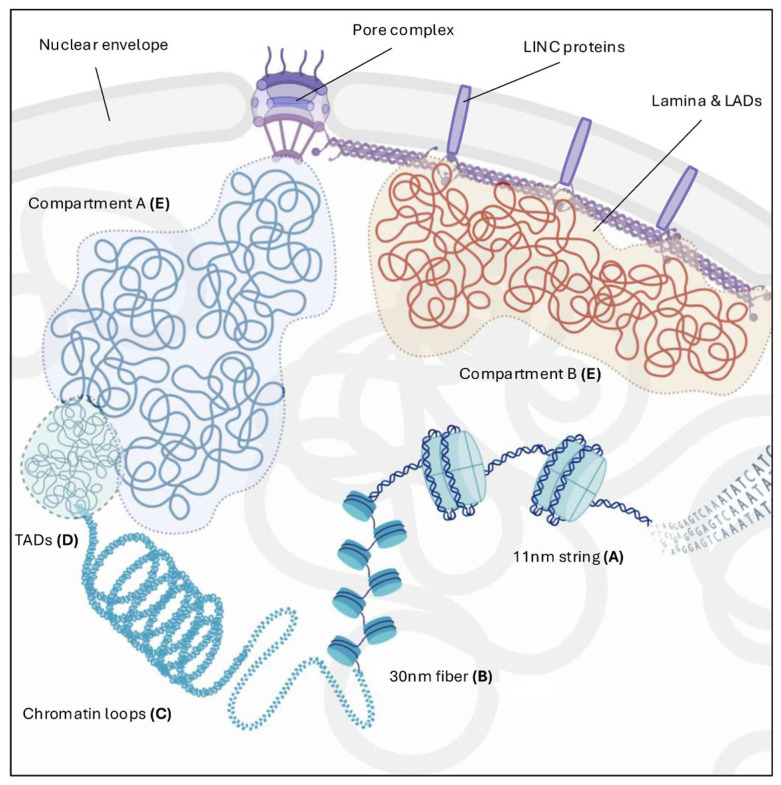
Illustration of chromatin organization, from basic to high-order levels: at first, DNA is wrapped around an octamer of histones (nucleosome) forming a 11 nm “beads-on-a-string” (**A**). Next, nucleosomes are packaged as a 30 nm fiber (**B**), which subsequently condensates and folds into chromatin loops (**C**). These are then arranged into topological associated domains (TADs) at a sub-megabase level (**D**), which interact between each other to create chromatin compartments (**E**). Chromatin alternates between compartments, in the form of euchromatin (comp-A), which has a more accessible state and gene activity, and heterochromatin (comp-B), which has a more dense and inactive conformation. Specifically, under the nuclear envelope, lamina-associated domains (LADs) are enriched of heterochromatin marks and in strict contact with lamins and LINC complex proteins. Created with BioRender.com (accessed on 7 March 2025).

**Figure 2 cancers-17-00949-f002:**
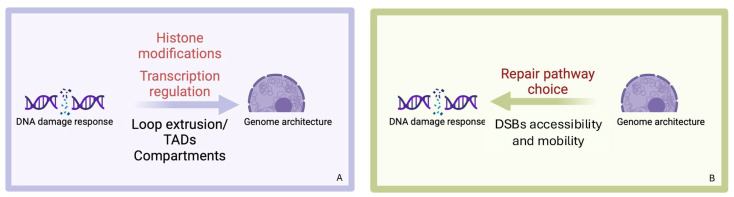
Following the formation of a DSB, nuclear architecture undergoes profound rearrangements to facilitate DNA repair. (**A**) At the local level, histone modifications, chromatin remodeling, and transcriptional regulation promote the *cis*-accumulation of repair factors. On a larger scale, inter-TAD and intra-TAD rearrangements enhance contacts in trans. (**B**) In turn, chromatin architecture can actively contribute to the DDR by influencing the repair pathway choice and increasing the accessibility and mobility of damaged loci, depending on the pre-existing chromatin context and/or the genomic location of DSBs. Created with BioRender.com (accessed on 7 March 2025).

**Figure 3 cancers-17-00949-f003:**
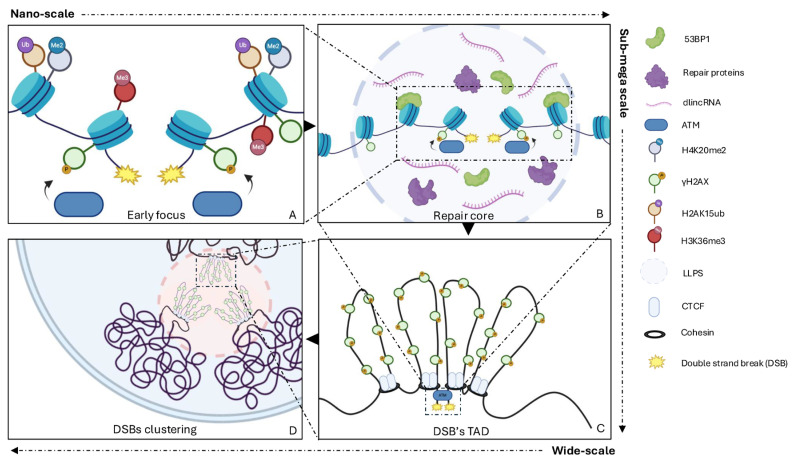
Stages of DSBs’ clustering. From nano-scale to sub-mega scale: (**A**) upon DSB, *cis* histone modifications are one of the earliest events at nano-scale level. (**B**) The formation of phase-separated repair cores, driven by the LLPS properties of 53BP1, facilitates the assembly of repair foci. From sub-mega scale to wide-scale: (**C**) repair foci form through the local rearrangement of chromatin at the level of TADs. The megabase-scale spreading of γH2AX is enabled by cohesin-mediated loop extrusion, which ensures ATM-dependent phosphorylation of H2AX across the entire TAD. (**D**) At a wider scale, γH2AX- and 53BP1-decorated damaged TADs can further fuse and self-segregate. This process relies on the actin network, the LINC complex embedded in the nuclear envelope, and PPPS properties of 53BP1. Created with BioRender.com (accessed on 7 March 2025).

**Figure 4 cancers-17-00949-f004:**
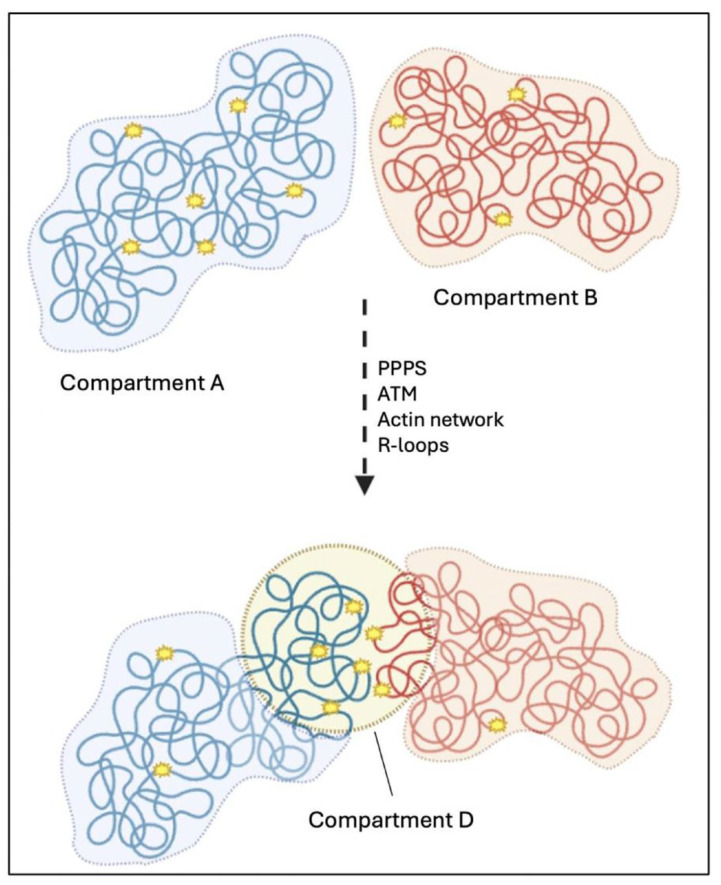
Schematic representation of D-compartments assembly following DSBs. Topologically associated domains (TADs) segregate into A and B compartments, in association respectively with active or inactive chromatin. Damaged TADs can form a specialized compartment, the ‘D compartment’, which is spatially isolated from the surrounding environment, through polymer–polymer phase separation (PPPS), ATM kinase activity, nuclear actin polymerization, and R-loop accumulation. Created with BioRender.com (accessed on 7 March 2025).

## Data Availability

Not applicable.

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
