# Peer review of "Interplay and Dynamics of Chromatin Architecture and DNA Damage Response: An Overview"

_cancers, 2025, doi:10.3390/cancers17060949_

Round 1

Reviewer 1 Report

Comments and Suggestions for Authors

Dear authors,

thank you for an interesting review on the relationship between DNA (DSB) repair and chromatin architecture at multiple levels of complexity. This relationship is more and more emerging and the complexity of processes regulated by chromatin structure during DSB repair just started to be disclosed. Therefore I consider the topic of the manuscript very important. As chromatin architecture/structure influences not only DSB repair but in principle all processes in the cell nucleus, I consider the article as interesting for a broad range of readers.

The manuscript is well written and organized. A positive point is that it covers the influence of chromatin architecture at multiple levels of complexity, starting with histone modifications up to organization of chromatin in the cell nucleus. Despite this broad range covered, the information provided is sufficiently detailed and allows to see not only chromatin functions at different levels of complexity, but also how they are mutually interconnected. Despite this in principle very positive impression, I have identified some minor but important points, which should be addressed before publishing. I only ask for minor revision.

  1. Page 3, line131: "HR is restricted to S/G2 phase only": As also stated in this manuscript later, HR can occur also in G1 phase, e.g., in form of RNA-mediated HR. So, I would modify sentence such as "HR is mostly restricted to S/G2 phase" (the references are provided later in this manuscript so it is not necessary to repeat them here, I think
  2. Page 3, line133: "if both these pathways are compromised, alternative end-joining...". In here, I understand what the authors mean but it is not clear how the word "compromised" should be understood - this can mean defects in both repair pathways as they occur, e.g., in cancer cells, but also point to the situation where both repair systems are overloaded with DSB damage, like after exposure to high doses of ionizing radiation. In both cases alternative repair pathways can be activated, which would be useful to explain in the text.
  3. Figure 1, right panel: Chromatin architecture influences DSB mobility as it is depicted in this figure panel. However, as also discussed by the authors later, chromatin architecture also influences the accessibility of DSBs for various repair proteins, which can be considered as a form of spatio-temporal regulation of the repair process. Therefore, I would add (below the arrow and "DSB mobility") also something like "DSB accessibility".
  4. Page 4, line170: Why a higher number of H4K16as motives is supposed to preferentially lead to HR? While H4K16ac, as stated some lines above, attracts 53BP1 to DSB, the decision for NHEJ or HR is made later, mediated at least partly by the mutual interaction between 53BP1 a BRACA1 complexes. Please explain (perhaps I have misinterpreted something)
  5. Similarly, Page 5, line179/180: "H4K20me1/2 decides if 53BP1 or BRACA1 is loaded" - as soon as I know, 53BP1 binds to in principle all gH2AX foci and later, mutual reorganization of 53BP1 and BRACA1 at the DSB site decides whether repair will proceed as NHEJ or HR (??)
  6. Page 7, para 2: This part of the text is not clear me - "Reduced number of CTCF results in lower numbers of smaller gH2AX foci. As CTCF makes barriers to spreading of gH2AX, I would expect their removal leading to perhaps less but larger gH2AX foci (due to "fusions" of otherwise separated gH2AX-covered domains (??)
  7. Page 7, 2nd para from the bottom: The relationship between heterochromatin or euchromatin or other chromatin structures remain highly controversial with references providing contradictory results. I think this uncertainty has to be emphasized here and more references, e.g. supporting HR, NHEJ or alternative pathways (e.g. Illiakis group papers) in heterochromatin discussed. Also see a review Falk and Hausmann, Cancers https://doi.org/10.3390/cancers13010018)
  8. Page 8, line 328-330: Indeed, there can be more mechanisms of clustering. As described in Falk et al, BBA MCR 2007, https://doi.org/10.1016/j.bbamcr.2007.07.002, heterochromatin has to decondense in order to allow repair to continue. This decondensation is associated with "protrusion" of heterochromatic DSBs to "euchromatic (ec) domains with low-dense chromatin". This mobilization of hcDSBs can occasionally lead to clustering of two or more DSBs (either hcDSB+ecEDS or hcDSB+hcDSB). These clusters can be considered as the byproducts of DSB repair and origins of a risk of chromosomal translocations, as also discussed later in this manuscript. These observations are well compatible with later cited Goutier group results pointing to compartment B to compartment A switch (in terms of Falk et al 2007, this corresponds to heterochromatin (compartment B) decondensation (structurally but nut functionally changing to compartment A) and consequent interactions between compartments B and A.
  9. Page 8, line 367: exactly this idea has been already proposed in 2007 in the work Falk et al BBA MCR 2007 mentioned in my previous point (8) (original paper) or reviewed in the context of formation of chromosomal translocations in Falk et al, doi: 10.1016/j.mrrev.2010.01.013.
  10. It is upon the consideration of the authors, but I would appreciate an additional image explaining the "clustering phenomenon" in DSB repair and potentially also a figure summarizing what chromatin architecture effects occur at different levels of chromatin organization (histone modification, ..., organization of chromatin in the cell nucleus), and how they are potentially interconnected.

Reviewer 2 Report

Comments and Suggestions for Authors

Review

This Review provides a comprehensive analysis of how chromatin architecture influences the DNA damage response (DDR), particularly in the context of double-strand breaks (DSBs). It explores the molecular mechanisms linking chromatin remodeling, nuclear organization, and DDR pathways, with an emphasis on how spatial genome organization contributes to genome stability and adaptability.

This Review is astonishingly interesting to read. It provides an in-depth exploration of the dynamic interplay between chromatin architecture and DDR. Despite its complexity, the Review serves as an essential resource for researchers investigating chromatin dynamics, DNA repair, and potential therapeutic interventions. The Review can be published in Cancers because it definitely will attract a lot of citations.  I would like to thank the Authors for well-polished writing

There are however several points:

Abstract:

  1. Please, outline clearly the aim of the review early in the text. 2. In the end of the Abstract, please, highlight the fundamental novelty of your Review compared to previous ones by emphasizing your analytical conclusions. Please, strengthen the importance of the role of chromatin regulation and DDR in the development of new therapeutic approaches for cancer therapy.
  2. Chapter 2 : Chromatin architecture: from nucleosome to higher order organization. Could you please provide more details about the "role of lamina and its partners in duplication, transcription and DNA damage repair factors as their direct mutation, dysregulation or incorrect post-translational maturation can alter any of these processes leading to various diseases [19] (Lines 100-102). Thank you for drawing attention to these little-studied properties of lamina. This information can enhance the novelty of the study.
  3. It also would be good to provide evidence about the effects of polyploidy on chromatin architecture and DDR. It was recently shown that polyploidy promotes chromatin opening via chromatin acetylation triggered by DDR activation. This context of polyploidy is also quite novel.
  4. Please, provide references for the text at lines 200-207.
  5. Please, prepare a small chapter or extend current fragments of the text concerning the role of DDR components in cancer therapies. Please, add something else besides PARP1
  6. It would be good to prepare figures clarifying Chapter 2
  7. Please, strengthen Conclusion by underlining novelty and therapeutic significance of the Review

Reviewer 3 Report

Comments and Suggestions for Authors

Ambrosio et al. present a review of the role of chromatin modifications and rearrangements in regulating the repair of DNA double strand breaks.  The article is clearly written and provides a useful compilation of the relevant primary research articles.   There are a small number of issues that the authors should consider.

1)  The title of the review incorporates the “DNA Damage Response” (DDR), a term that is generally understood to refer to signalling cascades that are activated by various forms of DNA damage or replication stress to trigger numerous downstream responses such as cell cycle arrest and inhibition of DNA replication (the ATM-Chk2 and ATR-Chk1 pathways).   The problem here is that, with the exception of ATM, the action of these pathways in chromatin regulation is really not discussed.  The focus is on the interaction of DNA repair processes and proteins with chromatin modification and rearrangement. A more precise title is therefore desirable to avoid misleading potential readers – one possibility would be “Interplay and Dynamics of Chromatin Architecture and DNA Repair”.  Obviously any change could require minor changes at various places in the text and figures.

2) Figure 2 presents a diagrammatic view of chromatin components in DNA repair foci at various “magnifications”, yet perplexingly these go from highest to lowest from left to right.  This is likely to be counter-intuitive for a majority of readers and the figure would be improved if the panels were presented in the opposite order; ie lowest to highest resolution from left to right.

Minor points

Line 131 – “restrained” – constrained would be more appropriate here.

Line 295  - “after DNA”. – something missing here?  Maybe repair?

Line 303 - “interacts” – interact

Lines 335-340 – The A, B, D chromatin groups need to be described here in a little more detail so readers can understand their significance without the need to consult the primary research papers in which they are defined.

Line 378 – “The findings” – These findings.

Lines 413-414 – “random “poking” of the nucleus in response to DNA” – not clear at all what is meant by this phrase.

Section 3.2.4 – the terms LINC complex, SUN and KASH domains, LADs and NADs should be defined on first usage as with other abbreviations .  
